# Parecoxib and 5-Fluorouracil Synergistically Inhibit EMT and Subsequent Metastasis in Colorectal Cancer by Targeting PI3K/Akt/NF-κB Signaling

**DOI:** 10.3390/biomedicines12071526

**Published:** 2024-07-09

**Authors:** Wan-Ling Chang, Jyun-Yu Peng, Chain-Lang Hong, Pei-Ching Li, Fung-Jou Lu, Ching-Hsein Chen

**Affiliations:** 1Department of Anesthesiology, Chang Gung Memorial Hospital at Chiayi, No. 8, West Section of Jiapu Road, Puzi City 613016, Chiayi County, Taiwan; chjack1975@yahoo.com.tw (W.-L.C.); pengjyun550@gmail.com (J.-Y.P.); leisure@cgmh.org.tw (C.-L.H.); peiching@cgmh.org.tw (P.-C.L.); 2Institute of Medicine, Chung Shan Medical University, No. 110, Section 1, Jianguo North Road, Taichung City 402306, Taiwan; fjlu@csmu.edu.tw; 3Department of Microbiology, Immunology and Biopharmaceuticals, College of Life Sciences, A25-303 Room, Life Sciences Hall, No. 300, Syuefu Road, National Chiayi University, Chiayi City 600355, Taiwan

**Keywords:** parecoxib, 5-fluorouracil, colorectal cancer, metastasis, reactive oxygen species, PI3K/Akt pathway

## Abstract

Colorectal cancer is one of the most common causes of cancer mortality worldwide, and innovative drugs for the treatment of colorectal cancer are continually being developed. 5-Fluorouracil (5-FU) is a common clinical chemotherapeutic drug. Acquired resistance to 5-FU is a clinical challenge in colorectal cancer treatment. Parecoxib is a selective COX-2-specific inhibitor that was demonstrated to inhibit metastasis in colorectal cancers in our previous study. This study aimed to investigate the synergistic antimetastatic activities of parecoxib to 5-FU in human colorectal cancer cells and determine the underlying mechanisms. Parecoxib and 5-FU synergistically suppressed metastasis in colorectal cancer cells. Treatment with the parecoxib/5-FU combination induced an increase in E-cadherin and decrease in β-catenin expression. The parecoxib/5-FU combination inhibited MMP-9 activity, and the NF-κB pathway was suppressed as well. Mechanistic analysis denoted that the parecoxib/5-FU combination hindered the essential molecules of the PI3K/Akt route to obstruct metastatic colorectal cancer. Furthermore, the parecoxib/5-FU combination could inhibit reactive oxygen species. Our work showed the antimetastatic capacity of the parecoxib/5-FU combination for treating colorectal cancers via the targeting of the PI3K/Akt/NF-κB pathway.

## 1. Introduction

At present, colorectal cancer is the third most popular cancer and the third most frequent cause of cancer-related mortality in the USA. About nine out of ten people with colorectal cancer are diagnosed when they are aged fifty or older. The mortality incidence of colorectal cancer decreases for people older than fifty but increases for those younger than fifty [1]. Surgery is only the first line of treatment for non-metastatic colorectal cancer [2], whereas chemotherapy is adjuvant in colon and rectal cancer stages III and IV. In locally advanced rectal cancer and total neoadjuvant therapy (with radiotherapy), the chemotherapy used is neoadjuvant chemotherapy (with radiotherapy). Despite the progress made in colorectal cancer treatment, the survival rate in advanced stages remains low [3]. The main factor resulting in mortality from colorectal cancer is metastatic disease [4]. Liver metastasis is the main cause of mortality in individuals with colorectal cancer [5]. Almost 25% of patients with colorectal cancer are clinically identified with liver metastasis at the point of disease diagnosis, and about 50% of patients with colorectal cancer develop liver metastasis-related symptoms during disease development [6]. About 30–50% of patients with colorectal cancer undergo recurring liver metastases following radical resection, and 50% eventually succumb to the disease [7]. However, the outcomes of current therapeutic strategies, including surgery, radiotherapy, and chemotherapy, for patients with colorectal cancer and distant metastasis remain poor because of drug resistance.

5-Fluorouracil (5-FU) is an old clinical chemotherapeutic agent. It has been employed since its invention 70 years ago. 5-FU-established chemotherapy was created to treat a variety of cancers, including colorectal cancer [8,9]. With the foremost developments in novel cancer therapies, 5-FU remains the primary treatment choice, usually in stage III and IV colorectal cancer [9,10,11]. Unfortunately, the efficiency of 5-FU diminishes when patients develop tolerance, which results mostly from the development of resistance in cancer cells [12,13,14]. However, with the absence of feasible therapeutic formulas for the resistance of colorectal cells to 5-FU, the increase in the dose and application of other therapies are the only clinical approaches to prolonging a patient’s life by several months [15]. Therefore, novel therapies or agents for 5-FU-resistant cancers are important.

5-FU/capecitabin is a first-line drug used in colon cancer stage III or IV as systemic chemotherapy (adjuvant chemotherapy or a palliative treatment), utilized alone or in association with oxaliplatin [16]. 5-FU is administrated intravenously. Capecitabin is administrated per os. 5-FU/capecitabin can also be used in rectal cancer treatment as one of the drugs needed, being employed in neoadjuvant therapy, in total neoadjuvant therapy, or as an adjuvant therapy, whether alone or in association with others [17]. Approximately 40% of colorectal cancer patients treated with 5-FU do not respond to the treatment or develop resistance. In such cases, 5-FU/capecitabin and oxaliplatin, which have a synergistic effect, are crucial. However, the use of oxaliplatin is associated with several side effects [18]. If a cyclooxygenase-2 (COX-2) inhibitor could replace oxaliplatin, it would significantly advance colorectal cancer treatment.

NSAIDs are inhibitors of the cyclooxygenase enzyme family, which catalyzes the metabolism of arachidonic acid into prostaglandins, prostacyclin, and thromboxane [19]. The COX-2 isoform is induced in response to cytokines and growth factors and is expressed in inflammatory disease, premalignant lesions (such as colorectal adenomas), and colon cancer [20]. Celecoxib, a COX-2 inhibitor drug, can be used in clinical practice in patients, treating familial adenomatous polyposis and patients with extent colorectal polyposis [21]. Parecoxib is also a COX-2 inhibitor but is used intravenously.

Parecoxib is a selective COX-2 inhibitor used for parenteral administration in clinical settings [22]. A previous study showed that parecoxib can remarkably increase radiation sensitivity in colorectal cells by directly influencing cancer cells and indirectly influencing the tumor vascular system [23]. Parecoxib is an enhancer of radiation therapy for colorectal cancer [23]. Our recent study demonstrated that parecoxib can inhibit metastasis in human colorectal cancers. Our findings showed a novel mechanism, underlying the antimetastatic activity of parecoxib in colorectal cancer by inhibiting p-Akt, p-ERK, p-GSK3β, MACC1, and cMet to induce the inhibition of epithelial–mesenchymal transition (EMT) and β-catenin [24]. These results illustrated that parecoxib may have novel potential activity, capable of enhancing clinical chemotherapeutic drugs for colorectal cancer treatment.

Reactive oxygen species (ROS) are important factors in cancer metastasis [25]. High amounts of ROS provoke cancer metastasis by stimulating the PI3K/Akt/mTOR and MAPK signaling pathways [26,27]. These signaling pathways activate downstream MMP-9, thereby initiating EMT, resulting in metastasis [28]. EMT plays a major role in metastasis; the polarity of epithelial cells leads to the disconnection of cell–cell adhesion and enhanced mobility [29]. Some studies have demonstrated that ROS is a critical factor that induces EMT [30,31]. These findings suggest that a certain generation of intracellular ROS can induce metastasis in cancer cells. Conversely, reducing ROS in cancer cells may inhibit metastasis.

Given that parecoxib is known to have potential anticancer activity [24] and 5-FU can synergistically enhance anti-tumor activity by aspirin, a NSAID [32], we wondered whether parecoxib may enhance 5-FU to inhibit metastasis in human colorectal cancer. To explore this possibility, we investigated whether the parecoxib/5-FU combination exhibits a superior ability to modulate the PI3K/Akt/NF-κB signaling pathways for promoting antimetastatic activity in human colorectal cancer. Moreover, we aimed to determine if the combination of parecoxib/5-FU can regulate intracellular ROS in human colorectal cancer cells. The relevant mechanisms behind the impact of the parecoxib/5-FU combination on antimetastatic effects will be discussed in this study.

## 2. Materials and Methods

### 2.1. Cell Line, Reagents, and Chemicals

The human colon cancer DLD-1 cell line (RRID: CVCL_0248) and SW480 cell line (RRID: CVCL_0546) were bought from the Bioresource Collection and Research Center (Hsinchu, Taiwan). An RPMI-1640 medium was purchased from Hyclone (South Logan, UT, USA). Fetal bovine serum (FBS) was obtained from Gibco Inc. (Freehold, NJ, USA). A protein assay Bio-Rad kit was from Bio-Rad Laboratories (Richmond, CA, USA). Parecoxib was obtained from Pfizer (Melbourne, Australia). Dimethyl sulfoxide (DMSO), 5-FU, trypan blue solution, crystal violet, 2′,7′-dichlorofluorescin diacetate (DCFH-DA), and other substances were purchased from Sigma-Aldrich Corp. (St. Louis, MO, USA).

### 2.2. Cell Culture and Drug Treatment

DLD-1 cells were incubated in RPMI-1640 (containing 10% FBS, 2 mM L-glutamine, 100 units/mL penicillin G, and 100 μg/mL streptomycin) and placed in an incubator below 5% CO_2_ at 37 °C. The stock solutions of parecoxib and 5-FU were made in DMSO, and total treatment concentrations were diluted in the incubated medium. The concentration of DMSO did not exceed 0.05%.

### 2.3. Cell Viability Assay

Cell viability was examined by the MTT assay. DLD-1 cancer cells (1 × 10^5^) were cultured in 24-well culture plates with a density of 1 × 105 cells/well for 24 h. The incubation medium was substituted with a fresh preparation, and the cells were incubated for 48 h with 0, 1, 3, 5, 7.5, and 10 μM parecoxib; 0, 10, 15, 20, 25, and 30 μM 5-FU; or parecoxib (3 μM) combined with 5-FU (15 and 20 μM). After incubation, cells were exposed to 0.5 mg/mL MTT solution for 2 h and dissolved with DMSO. An aliquot (200 μL) of the DMSO-dissolved solution was obtained from 12-well culture plates and transported to 96-well ELISA plates. The absorbance of the DMSO-dissolved suspension in the 96-well ELISA plates was detected at 595 nm in a microplate reader (Bio-Rad, Richmond, CA, USA).

### 2.4. Transwell Migration and Transwell Invasion Assay

Assays were implemented, using 24-well Millicell^®^ (Merck, Taipei City, Taiwan) chamber transwell plates for a migration assay and 24-well Matrigel invasion chambers for an invasion assay with a polyethylene terephthalate membrane. The DLD-1 cells (1 × 10^5^ cells/well) or SW480 cells (7 × 10^4^ cells/well) were placed in upper-chamber inserts with a serum-free medium, and the bottom chamber included a whole medium containing 10% FBS. DLD-1 cells were treated with parecoxib (3 μM) alone, 5-FU (15 or 20 μM) alone, parecoxib (3 μM) combined with 5-FU (20 μM), and parecoxib (3 μM) combined with 5-FU (20 μM) after being co-treated with wortmannin (20 μM) for 48 h. SW480 cells were treated with parecoxib (2 or 3 μM) alone, 5-FU (3 or 5 μM) alone, and parecoxib (3 μM) combined with 5-FU (5 μM) for 48 h (migration assay) or 96 h (invasion assay). After drug treatment, the upper-chamber inserts were swabbed to eliminate the non-travelling cells, and the travelling cells that were distributed across the polyethylene terephthalate film were fixed with 4% formaldehyde and stained with 0.05% crystal violet solution. Cells that passed through the pores were calculated in three randomly selected areas under a phase-contrast microscope. Each chamber insert was transferred to an unused well including 200 μL of the dissolving solution (33% acetic acid) to dissolve the cells, and 100 μL of each dissolving solution was transferred and assessed at 570 nm using an EnSpire^®^ multimode plate reader (PerkinElmer, Waltham, MA, USA).

### 2.5. Isobologram Analysis

Isobologram analysis was conducted to identify synergistic antimetastatic effects following treatment with parecoxib and 5-FU. When two drugs are used in treatment concomitantly, the antimetastatic effect of the combination must be distinguished from the antimetastatic effect of each drug. Treatment with more than one drug may lead to effects that are less than, or greater than, the additive effect of each drug when given alone. An isobologram is a non-mechanistic method of explaining the effect resulting from treatment with two drugs. An isobologram offers a suitable graphical demonstration of individual concentrations through rectangular coordinates (x, y) in order to illustrate equieffective pairs of dosages of drugs X and Y, respectively. Therefore, the isobologram involves a set of noticeable, experimentally defined points whose position allows a category of drug–drug interaction. For drug alone treatment, parecoxib alone (1, 2, and 3 μM) and 5-FU alone (5, 10, and 20 μM) were selected. For drug combination, 1 μM parecoxib + 5 μM5-FU, 2 μM parecoxib + 10 μM 5-FU, and 3 μM parecoxib + 20 μM 5-FU were selected. After 48 h of drug treatment, transwell migration was evaluated, and isobologram analysis of parecoxib combined with 5-FU in the DLD-1 cell line was performed using CalcuSyn software 1.0. Combination index data were computed using CalcuSyn software 1.0. The points below the backslash line indicate the synergistic outcome. The combination index (CI) was then assessed.

### 2.6. Gap Closure Assay

The migration capacity of colon cancer cells was determined via a gap closure assay. Four-well culture inserts were fixed onto six-well plates. DLD-1 cells (4.4 × 10^4^) were seeded onto each 4-well culture insert in 6-well plates for 24 h, and all 4-well culture inserts were removed to create wounded regions. Subsequently, the plates were replenished with a fresh culture medium, containing concentrations of parecoxib (3 μM) alone, 5-FU (20 μM) alone, and a parecoxib (3 μM)/5-FU (20 μM) combination. The plates were incubated for 24 and 48 h. The number of migrating cells was detected and counted using a light microscope. The Image J software 1.49 was used to analyze cell migration into cross-shaped areas.

### 2.7. Western Blot Analysis

DLD-1 cells were seeded in 6 cm dishes with a density of 4 × 10^5^ cells/well. We added arecoxib (3 μM) alone, 5-FU (20 μM) alone, and parecoxib (3 μM) combined with 5-FU (20 μM), and the cells were incubated for 24 and 48 h. After incubation with the indicated drugs for various time points, all cells were cleaned with PBS, refloated in a protein extraction solution for 10 min, and centrifuged at 12,000× *g* for 10 min at 4 °C to extract proteins. The protein concentrations were quantified with a Bio-Rad protein analysis kit. The cellular extracted proteins were heated in a loading buffer, and an equal quantity of protein (30–50 μg) was divided on a 12% SDS–polyacrylamide gel and then transferred to PVDF membranes. The membranes were immersed with a number of primary antibodies overnight and washed with a PBST solution containing 0.05% Tween 20 PBS/PBS. After cleaning, the suitable secondary antibodies containing horseradish peroxidase were added to the membrane for 1 h, which was cleaned with a PBST solution containing 0.05% Tween 20/PBS. The antigen–antibody complexes were recognized via enhanced chemiluminescence (Amersham Pharmacia Biotech, Piscataway, NJ, USA) with a chemiluminescence analyzer.

### 2.8. Antibodies

Antibodies against E-cadherin (cat. no. sc-7870), β-catenin (cat. no. sc-7963), p-Akt (Ser473; cat. no. sc7985-R), p-IκB (cat. no. sc-8404), p-Akt (cat. no. sc-7985-R), tubulin (cat. no. sc-5286), vimentin (cat. no. sc-6260), actin (cat. no. sc-1616-R) total-Akt (cat. no. sc-8312), total-IKK (cat. no. sc-7606), total-IκB (cat. no. sc-371), total-p65 (cat. no. sc-372), and GAPDH (cat. no. sc-47724) were purchased from Santa Cruz Biotechnology, Inc. (Santa Cruz, CA, USA). p-IKK (cat. no. ab138426), Histone H3 (cat. no. ab1791), and p-p65 (cat. no. ab28856) were bought from Abcam (Waltham, MA, USA). The anti-rabbit IgG (cat. no. ab150077) and anti-mouse IgG (cat. no. ab6708) secondary antibodies were purchased from Abcam (Waltham, MA, USA).

### 2.9. Gelatin Zymography Analysis

Gelatin zymography was carried out to verify the activity of MMP-9 in the media. DLD-1 cells were seeded in 6 cm dishes with a density of 4 × 10^5^ cells/well. We added parecoxib (3 μM) alone, 5-FU (20 μM) alone, and parecoxib (3 μM) combined with 5-FU (20 μM), and the cells were cultured for 24 and 48 h. After incubation, a medium containing 40 μg of protein was produced using an SDS sample solution without heating or a reducing agent and exposed to 0.2% gelatin in 10% SDS–PAGE electrophoresis. After electrophoresis, the gels were cleaned with 2.5% Triton X-100 and immersed in a reaction solution (5 mM CaCl_2_; 1 μM ZnCl_2_; 0.02% NaN_3_ and 50 mM Tris–HCl, pH 7.4) at 37 °C for 20 h. The gel was dyed with Coomassie brilliant blue R-250 for imaging. Data normalization was performed via Image J v1.46.

### 2.10. Effect of p-Akt Overexpression

About 6 μg of myc-tagged myristoylated Akt expression vector (Myr-Akt, Addgene, Cambridge, MA, USA) or empty vector (pUSEamp, Upstate Technology, Skaneateles, NY, USA) was transfected to DLD-1 cells by Lipofectamine (Invitrogen Taiwan Ltd., Kaohsiung, Taiwan), following the manufacturer’ s method. The G418 selection solution (400 μg/mL) was used to maintain a stable cell line for 14 days. The stable cell line was incubated in an RPMI-1640 medium including 10% FBS and then subjected to the parecoxib (3 μM)/5-FU (20 μM) combination for 48 h. After drug incubation, the cells were used for migration examination.

### 2.11. Intracellular ROS Analysis

The generation of intracellular ROS was assessed by flow cytometry performed using DCFH–DA. DLD-1 cells were seeded in 6 cm dishes with a density of 4 × 10^5^ cells/dish. Parecoxib (3 μM) alone, 5-FU (20 μM) alone, and parecoxib (3 μM) combined with 5-FU (20 μM) were added, and the cells were treated for 6 h. After drug incubation, the cells were dyed with 20 μM DCFH–DA for 30 min at 37 °C and then cleaned with 1 × PBS twice to eliminate DCFH–DA. All cells were trypsinized to gain a single-cell suspension. Using a BD FACSCantoTM II flow cytometer (San Jose, CA, USA), we measured intracellular ROS amounts, which were detected by the fluorescence of dichlorofluorescein (DCF), through excitation/emission: 485 nm/535 nm H_2_O_2_ (800 μM) was used as an intracellular ROS positive control. About 10,000 cells were collected and analyzed, as per experimental conditions, via mean fluorescent intensity assessments.

### 2.12. Statistical Analysis

Data are presented as the mean ± standard deviation from at least three independent experiments and analyzed using Student’s *t* tests. A *p* value of less than 0.05 was considered statistically significant.

## 3. Results

### 3.1. Cell Viability on Parecoxib and 5-FU Treatment

The concentrations of parecoxib and 5-FU in DLD-1 cells must have low cytotoxicity in order to evaluate the anti-migration effect of parecoxib combined with 5-FU. This will not affect the evaluation of anti-migration effects due to excess cytotoxicity. As shown in Figure 1, all concentrations of parecoxib alone, 5-FU alone, and parecoxib combined with 5-FU resulted in cell viabilities that exceeded 80%. We selected 3 μM parecoxib combined with 20 μM 5-FU for use in further experiments in order to evaluate the mechanism underlying the anti-migration effect.

### 3.2. Synergism of Parecoxib and 5-FU Combination in Antimetastasis in DLD-1 Cells

We first used transwell assays to evaluate whether parecoxib can enhance the anti-migration effects of 5-FU. Treatment with 5-FU alone inhibited migration to 70%. However, we noticed a significant inhibition (50%) in cell migration after combination treatment with 3 μM parecoxib/20 μM 5-FU (Figure 2A,B). Synergism was studied via isobologram analysis to determine the synergistic effect of the parecoxib/5-FU combination on inhibiting migration. The synergism of the parecoxib/5-FU combination in terms of preventing migration was determined by isobologram analysis, which was performed to analyze the interaction of two treatments. As shown in Figure 2C, 1, 2, and 3 μM parecoxib were combined with 5, 10, and 20 μM 5-FU, respectively, resulting in the synergistic effect of inhibiting cell migration in DLD-1 cells. We further loaded Matrigel in transwell assays to evaluate whether parecoxib can enhance the anti-invasion ability of 5-FU. As shown in Figure 2D,E, the use of 3 μM parecoxib alone and 20 μM 5-FU alone decreased cell invasion to 78% and 79%, respectively, whereas the parecoxib/5-FU combination significantly decreased invasion to 47%; these results indicated that the parecoxib/5-FU combination synergistically inhibited metastasis.

### 3.3. Parecoxib Enhances Gap Closure Inhibition of 5-FU Treatment in DLD-1 Cells

To further verify whether horizontal migration is enhanced by parecoxib in 5-FU-treated cells, we evaluated the horizontal migration ability of cells using gap closure assays. The results indicated that gap closure significantly decreased in the parecoxib/5-FU combination compared with that seen in parecoxib-alone and 5-FU-alone treatment (Figure 3A,B).

### 3.4. EMT Suppression on Parecoxib and 5-FU Combination

In colorectal cancer, EMT is associated with an invasive or metastatic phenotype. We further evaluated the effect of the parecoxib/5-FU combination on EMT. The epithelial cell marker E-cadherin and mesenchymal cell marker β-catenin were assessed via Western blot analysis. As shown in Figure 4, the expression of E-cadherin increased in the parecoxib/5-FU combination compared with that seen in parecoxib alone and 5-FU alone at 48 h. Conversely, the parecoxib/5-FU combination inhibited the expression levels of β-catenin, fibronectin, Snail, and vimentin more than parecoxib alone and 5-FU alone at 24 and 48 h. These results indicated that the parecoxib/5-FU combination reversed EMT in DLD-1 cells.

### 3.5. Inhibition of MMP-9 Activity on Parecoxib and 5-FU Combination

MMP-9 has been implicated in the progression and metastasis of various cancers. The activity of MMP-9 in parecoxib/5-FU combination treatment was analyzed via zymography. As shown in Figure 5, we found that using parecoxib alone and 5-FU alone could slightly inhibit the activity of MMP-9. Notably, the parecoxib/5-FU combination inhibited the activity of MMP-9 more than using parecoxib alone and 5-FU alone at 24 and 48 h.

### 3.6. Inhibition of PI3K/Akt/NF-κB Pathway on Parecoxib and 5-FU Combination

The PI3K/Akt pathway mediates EMT in many cancer cells. NF-κB is a downstream component of the PI3K/Akt pathway. Therefore, we assessed Akt phosphorylation (p-Akt) via Western blot analysis in DLD-1 cells treated with the parecoxib/5-FU combination. p-Akt expression was significantly reduced in the parecoxib/5-FU combination compared with that in treatment with parecoxib alone and 5-FU alone at 48 h (Figure 6A). The use of parecoxib alone did not inhibit the expression levels of p-IKK and p-p65. Notably, the parecoxib/5-FU combination enhanced the expression levels of p-IKK and p-p65 compared with 5-FU-alone treatment (Figure 6B). These results indicated that parecoxib enhanced the inhibitory effect of 5-FU on the PI3K/Akt/NF-κB pathway.

### 3.7. Akt Phosphorylation as the Critical Mediator by Which Parecoxib Can Potentiate the Anti-Migration Ability of 5-FU

To further confirm the anti-migration role of p-Akt inhibition in parecoxib/5-FU treatment in DLD-1 cells, we used the p-Akt overexpression vector to enforce exogenous p-Akt expression. As shown in Figure 7A, the overexpression of p-Akt could increase p-Akt (Ser 473) expression 1.59-fold compared with that of DLD-1 cells. The overexpression of p-Akt effectively reversed the inhibition of migration by the parecoxib/5-FU combination in DLD-1 cells (Figure 7B,C), indicating that p-Akt is a critical mediator in the parecoxib/5-FU combination. Furthermore, we used wortmannin, an Akt inhibitor, to inhibit the phosphorylation of Akt. Our results demonstrated that wortmannin suppressed the phosphorylation of Akt (Figure 7D) and significantly inhibited the migration and invasion of cells treated with the parecoxib and 5-FU combination (Figure 7E,F).

### 3.8. Parecoxib/5-FU Combination Inhibits Intracellular ROS

To explore the possible role of ROS in the parecoxib/5-FU combination in DLD-1 cells, we performed DCFH-DA staining and flow cytometry to evaluate the intracellular ROS levels. As shown in Figure 8, the use of parecoxib alone and 5-FU alone did not affect intracellular ROS compared with untreated cells. Compared with cells treated with parecoxib alone and 5-FU alone, the intracellular ROS levels decreased significantly in the parecoxib/5-FU combination.

### 3.9. Synergism of Parecoxib and 5-FU Combination in Antimetastatic Effect in SW480 Cells

To evaluate the synergistic effects of the parecoxib and 5-FU combination on another human colorectal cancer cell line, we used the SW480 cell line, transwell migration assay, and Matrigel invasion assay. Treatment with 5-FU alone inhibited migration to 71%. However, we found a significant inhibition (57%) in cell migration after the 3 μM parecoxib/5 μM 5-FU combination treatment (Figure 9A,B). The synergistic effect of inhibiting cell migration in the parecoxib/5FU combination was demonstrated in SW480 cells (Figure 9C). Moreover, the anti-invasion experiments demonstrated that the parecoxib/5-FU combination significantly decreased invasion compared with parecoxib-alone and 5-FU-alone treatments (Figure 9D,E).

## 4. Discussion

5-FU represents an anti-metabolite that replaces fluorine for hydrogen at the C-5 position of uracil. The thymine–uracil/5-FU exchange produced by the thymine substitute in DNA therefore results in the formation of adenine–uracil/5-FU base pairs [33]. The mechanism behind the antitumor effects of 5-FU chiefly occurs via the repression of thymidylate synthase, leading to the disturbance of the intracellular deoxynucleotide pools needed for DNA replication [33]. Our previous study demonstrated that the parecoxib antimetastasis mechanism effect is correlated with the attenuated phosphorylation of Akt, the expression of vimentin (a mesenchymal marker), and β-catenin, and corresponds with the up-regulated GSK3β and E-cadherin (an epithelial marker) in human colon cancer cells [24]. Recent studies have demonstrated that Akt/NF-κB signaling is strongly associated with metastasis in colorectal tumor cells [34,35]. Parecoxib was demonstrated by us to inhibit the Akt phosphorylation and EMT in human colorectal cancer at clinical concentrations [24]. 5-FU has the ability to inhibit the migration and invasion of colorectal cancer cells via the PI3K/Akt pathway [36]. It is necessary to find a potential drug with which to enhance the antimetastatic ability of 5-FU. Thus, we speculated that in colorectal cancer cells, the anti-migration and anti-invasion efficacy of 5-FU might be increased in combination with parecoxib. Our results demonstrated that parecoxib enhanced the inhibition of 5-FU along the EMT (Figure 4) and PI3K/Akt/NF-κB pathways (Figure 6 and Figure 7) and subsequently enhanced the inhibition of MMP-9 activity (Figure 5). These results confirm our hypothesis that 5-FU and parecoxib exhibit an inhibitory synergistic effect on migration and invasion in colorectal cancer cells. The PI3K/AKT signaling pathway has been demonstrated to be a critical regulator of epithelial-mesenchymal transition in colorectal tumor cells [37]. There are many compounds and substances that can exhibit antimetastasis through inhibiting the PI3K/AKT signaling pathway and regulating EMT. For example, berberine inhibited the migration and invasion of some colon cancer cell lines by up-regulating PTEN, which repressed the PI3K/AKT pathway at the gene and protein levels [38]. *Antrodia camphorate* significantly inhibited migration and invasion, accompanied by the down-regulation of MMP-2 and MMP-9 proteins via the inhibition of the PI3K/AKT/NFκB signaling pathways [39]. *Astragalus mongholicus* Bunge-*Curcuma aromatica* Salisb. inhibited the expression and transcription of genes related to the PI3K/AKT pathway while inhibiting the EMT process in colon cancer cells and model mice [40]. The present findings of this research are consistent with these previous studies. The inhibition of PI3K/Akt/NF-κB pathway, EMT and MMP-9 activity are critical events in antimetastasis effect of 5-FU/parecoxib combination in colorectal cancer.

Our previous reports demonstrated that parecoxib can significantly repress the cell migration of DLD-1 and SW480 cells [24]. The mechanisms included the inhibition of metastasis-associated colon cancer 1 (MACC1), Akt phosphorylation, β-catenin expression, and EMT [24]. Other authors also revealed that parecoxib can suppress metastasis by reducing the expression of engulfment and cell motility 3 (ELMO3) in lung cancer [41]. These results illustrated that parecoxib may offer antimetastatic potential in combination with chemotherapeutic agents in cancer treatment. MACC1 has been demonstrated to play an important role in metastasis formation in colorectal cancer [42]. Knockdowns of MACC1 can facilitate cell death via the PI3K/Akt pathway in 5-FU-resistant colorectal cancer [43]. Many studies have demonstrated that MACC1 inhibition can enhance antimetastatic activity. Curcumin reduces MACC1 expression and inhibits wound healing in colorectal cancer [42]. Selumetinib treatment induces significant repression in metastasis, but only in MACC1-positive xenografts [44]. These results indicate that MACC1 is an antimetastatic target in colorectal cancer. Recently, Kortüm et al. demonstrated that the combination of statins and niclosamide prevents colorectal cancer propagation by opening the MACC1–β-catenin–S100A4 axis of metastasis [45]. In our previous study, we demonstrated that parecoxib inhibits the expression levels of MACC1. Thus, we speculated that the antimetastatic effect of 5-FU, enhanced by parecoxib, may partly result from the inhibition of MACC1.

Our present study revealed that 5-FU alone and parecoxib alone did not affect the intracellular ROS levels. Notably, the present study found that the 5-FU/parecoxib combination could significantly inhibit ROS in colorectal cancer cells for 48 h. Some reports indicated that ROS often plays a dual role in regulating various types of cellular metabolism [46,47]. The intracellular ROS concentration determines the direction of dual role. In general, cancer cells show higher basic concentrations of ROS compared with normal cells. At low to moderate concentrations, ROS function as signal transducers that trigger cell migration, invasion, proliferation, and angiogenesis. By contrast, high concentrations of ROS result in damage to membranes, lipids, proteins, nucleic acids, and organelles, causing cell death. ROS provoke tumor metastasis and invasion by prompting EMT in cancer cells. Notably, a large accumulation of ROS represses tumor growth through two mechanisms: (1) by blocking the signaling pathway responsible for cancer cell proliferation, as well as the cell cycle and the biosynthesis of ATP and nucleotides; and (2) by promoting tumor cell death via activating the p53-independent, mitochondrial, and endoplasmic reticulum stress–apoptotic pathways and the ferroptosis pathway [46]. As a result of the elevated levels of ROS in cancer cells, the administration of medications that further increase ROS concentrations can effectively impede the metastasis of cancer cells and even trigger apoptosis in these cells. These phenomena are consistent with the findings of Piskounova et al., who reported that ROS inhibit the spread of melanoma cancer in vivo [48]. However, some studies showed that reducing the ROS status in cancer cells will inhibit various activities of cancer cells, including the mechanism regulating metastasis [49,50]. Our previous studies also demonstrated that the inhibition of intracellular ROS can repress cancer metastasis [51]. We found that ROS inhibition by propyl gallate treatment can significantly inhibit metastasis in glioblastoma cells [51]. Another study illustrated that mitoQ, a mitochondria-targeted antioxidant, decreases mitochondrial superoxide production and inhibits metastasis in human pancreatic cancer [52]. Febuxostat, a xanthine oxidase inhibitor, alleviates breast cancer cell metastasis related to the inhibition of ROS [53]. We speculate that this negative regulation of ROS may be related to the inhibition of metastasis. However, the relationship between ROS inhibition induced by the 5-FU/parecoxib combination and metastasis in vivo must be further explored in the future. 5-FU can increase ROS in colorectal cancer cells [54]. Our present study reveals that 5-FU/parecoxib combination does not increase the intracellular ROS levels. Conversely, 5-FU/parecoxib combination inhibited the intracellular ROS levels. The result is unexpected. COX-2 is an important source of ROS generation [55]. COX-2-derived different prostanoids can modulate ROS production [55]. One possible reason is that parecoxib is an NSAID, which can specifically inhibit the production of ROS mediated by COX-2, resulting in the inability of 5-FU/parecoxib combination to induce an increase in ROS.

The previous literature has shown that 5-FU inhibits the migration and invasion of colorectal cancer cells through the PI3K/Akt pathway [36]. Similarly, we confirmed that 5-FU (15 μM and 20 μM) alone could inhibit about 30% of migration and that the inhibition did not proceed in a dose-dependent manner (Figure 2B). These results indicate that 5-FU alone can moderate inhibit the migration and invasion of colorectal cancer cells. Our present results show that parecoxib (3 μM) slightly increased the migration, but that it was not significant. In Figure 3B, the inhibition of migration of 5-FU/parecoxib combination is shown to be weaker at 48 h than at 24 h. It might be that the cancer cells were adapting to the crawling environment between 0 and 24 h, making the inhibitory effect of migration on the drugs more significant. However, during 24 to 48 h, some of the drugs might be metabolized, and so the inhibitory effect is relatively weak. In our previous studies, we demonstrated that 10 μM parecoxib inhibited the metastasis of colon cancer through the PI3K/Akt signaling pathway [24]. We hypothesized that low concentrations of parecoxib may enhance 5-FU’s ability to inhibit metastasis. As expected, the present results demonstrated that a low concentration of parecoxib (3 μM) and 5-FU synergistically inhibited the migration and invasion of DLD-1 cells. To further verify the relationship between the inhibition of the PI3K/Akt pathway and metastasis in the parecoxib/5-FU combination, we used the overexpression of p-Akt in DLD-1 cells in our experiments. The results indicated that migration significantly increased in the p-Akt-overexpressing DLD-1 cells under the parecoxib/5-FU combination. On the other hand, wortmannin co-treatment with parecoxib/5-FU combination further increased the anti-migration effect. Similarly, a previous study reported that the PI3K/Akt/NF-κB/MMP-9 signaling pathway plays an essential role in tumor metastasis and growth [56]. The activation of the PI3K/Akt signaling pathway can regulate cancer cell metastasis by increasing the expression of MMP-9 at the transcriptional level [57]. MMP-9 can destroy type IV collagen, which exists in the basement membrane of the extracellular matrix and elevates cancer metastasis [58,59]. PI3K/Akt can regulate the downstream of the NF-κB transcription factor to transcribe many metastasis-related genes, including the expression of MMP-9 and EMT markers, to promote the metastasis of cancer cells [60]. Our results showed that the parecoxib/5-FU combination led to the inhibition of downstream effectors such as the NF-κB transcription factor, EMT, and MMP-9 activity, indicating that the parecoxib/5-FU combination exerted antimetastatic activity by inhibiting the PI3K/Akt signaling pathway.

Cumulative reports have shown that the transcription factor Snail contributes to the conversion from epithelial-like cells into mesenchymal-like cells via repressing E-cadherin expression in many cancer cells [61]. To discover whether Snail is included with the EMT process, we detected the expression of Snail in DLD-1 cells treated by the parecoxib/5-FU combination. We found that the parecoxib/5-FU combination resulted in the inhibition of Snail to less than half in DLD-1 cells (Figure 4). As a transcriptional factor, Snail is triggered by the PI3K/Akt signaling pathway [62]. Moreover, our results showed that the expression of p-Akt was inhibited by the parecoxib/5-FU combination (Figure 7A). Thus, the EMT process, which was induced by the downstream effect of the PI3K/AKT pathway, was inhibited by the combination of parecoxib and 5-FU.

The weak point of the methodology is the lack of a PCR method with which to analyze various genes expression. However, PCR methods cannot evaluate the phosphorylation of Akt. The strong points of the methodology are to use the Western blot to analyze various protein expressions, including the phosphorylation of Akt, and to use the zymography to analyze the MMP-9 activity. Moreover, the overexpression of p-Akt, confirming the detailed antimetastasis mechanism in the 5-FU/parecoxib combination, is also a strong point of the methodology.

Both 5-FU and parecoxib are already commonly used drugs in clinical practice. The concentrations used in our study were also within the human serum concentration range. Animal experiments involving xenograft colorectal cancer should be planned in the future to validate the antimetastatic results obtained from our cell experiments in order to connect these basic scientific findings and clinical practice.

## 5. Conclusions

In this study, we demonstrated that parecoxib synergistically enhanced 5-FU to inhibit metastasis in DLD-1 cells. The antimetastatic mechanisms of the parecoxib/5-FU combination included (1) the inhibition of the PI3K/Akt/NF-κB signaling pathway; (2) the inhibition of MMP-9 activity; (3) the inhibition of EMT; and (4) the inhibition of intracellular ROS. The results showed that parecoxib cooperated with 5-FU to enhance the inhibition of colorectal cancer migration and invasion. Thus, the parecoxib/5-FU combination can contribute to the development of colorectal cancer treatment.

## Figures and Tables

**Figure 1 biomedicines-12-01526-f001:**
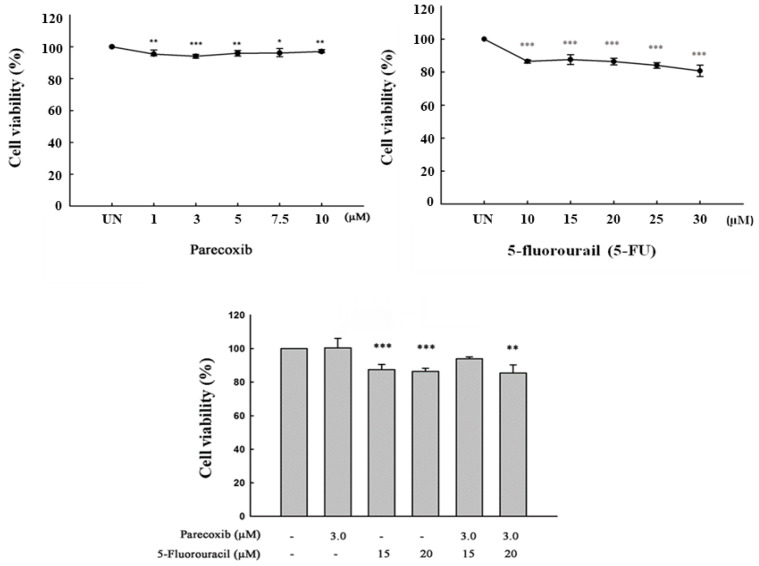
Effect of parecoxib and 5-FU on cell viability, as assessed by MTT assay. After incubation, cell viability was assessed by MTT analysis. Significant differences in the untreated group (UN) are shown as follows: *p* < 0.05 (*), *p* < 0.01 (**), and *p* < 0.001 (***).

**Figure 2 biomedicines-12-01526-f002:**
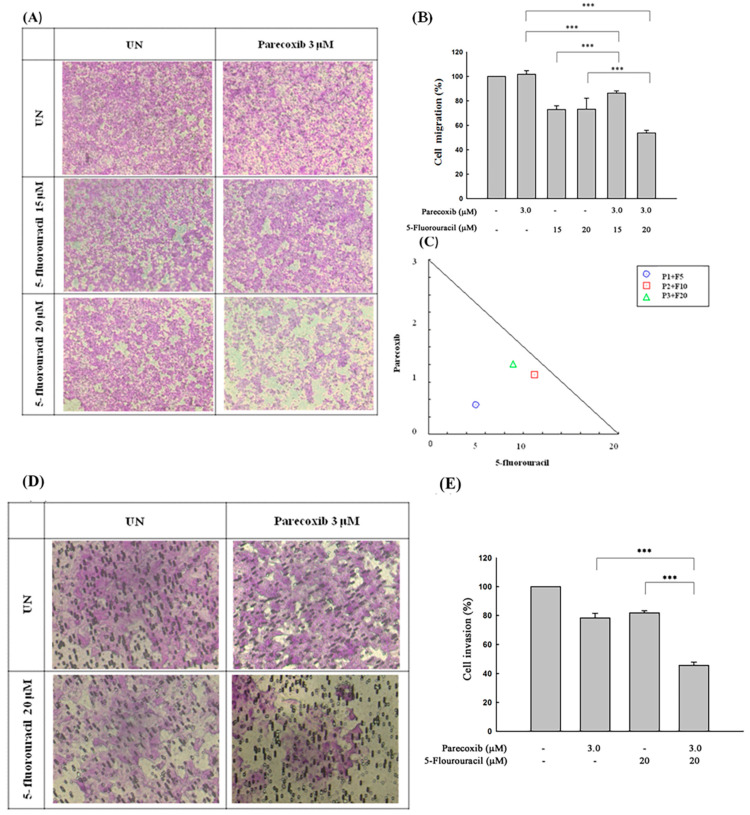
Effect of parecoxib and 5-FU on cell migration and invasion by transwell and matrix gel assays in DLD-1 cells. (**A**,**B**) Migration assay. (**D**,**E**) Invasion assay. (**A**,**D**) Arbitrary fields from each of the triplicate migration assays were calculated using a phase-contrast microscope (magnification 200×). (**B**,**E**) The absorbance of crystal violet was determined at 570 nm by a microplate reader. The values are displayed as the mean ± SD of separate trials. Significant differences are set at *p* < 0.001 (***). (**C**) Isobologram analysis of the parecoxib and 5-FU combination in DLD-1 cells. The trials were conducted at least three times. A descriptive trial is shown.

**Figure 3 biomedicines-12-01526-f003:**
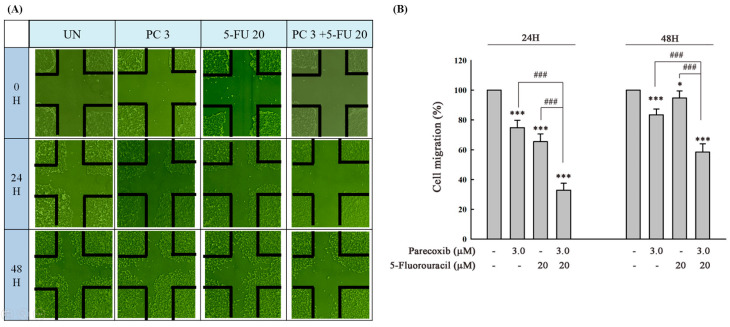
(**A**) After treatment, the cells that migrated to the wounded regions were calculated by a phase-contrast microscope (magnification 200×). (**B**) Percentages of DLD-1 cells that migrated to the wound area following treatment were evaluated. Significant differences in the untreated group (UN) are denoted as *p* < 0.001 (***). The * is a statistical symbol. Significant differences in parecoxib (3 μM) alone or 5-FU (20 μM) alone are indicated as *p* < 0.001 (###).

**Figure 4 biomedicines-12-01526-f004:**
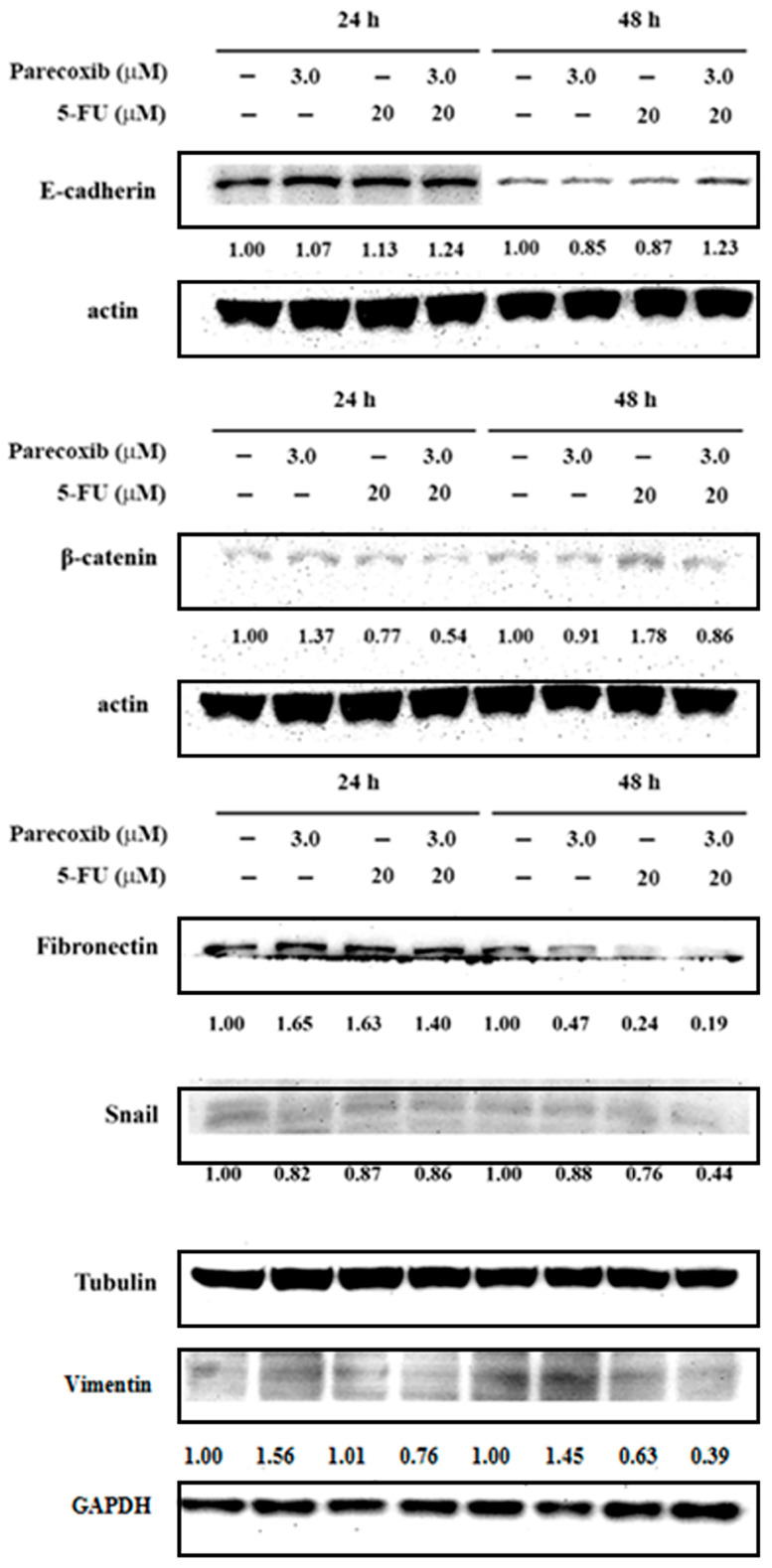
Effect of parecoxib and 5-FU on EMT. After treatment, the levels of protein expression were evaluated using the extracted proteins and assessed by Western blot. Actin, tubulin, or GAPDH were used as internal controls.

**Figure 5 biomedicines-12-01526-f005:**
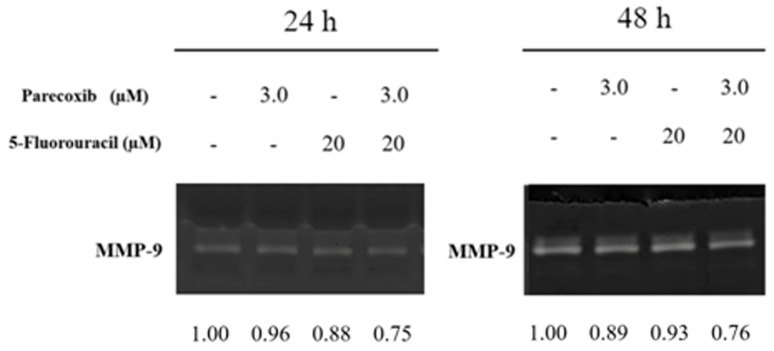
Effect of parecoxib and 5-FU on MMP-9 activity. After treatment, the conditional media were used on non-reduced denatured 12% polyacrylamide gel containing gelatin and stained with Coomassie Blue.

**Figure 6 biomedicines-12-01526-f006:**
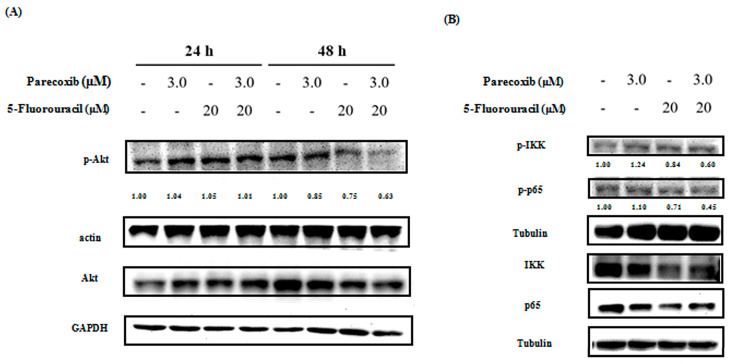
Effect of parecoxib and 5-FU on the p-Akt and NF-κB pathways. The cells were treated with drugs for (**A**) 24 and 48 h, and for (**B**) 1 h. After incubation, levels of protein expression were assessed via Western blot analysis. Actin, tubulin, or GAPDH were selected as loading controls.

**Figure 7 biomedicines-12-01526-f007:**
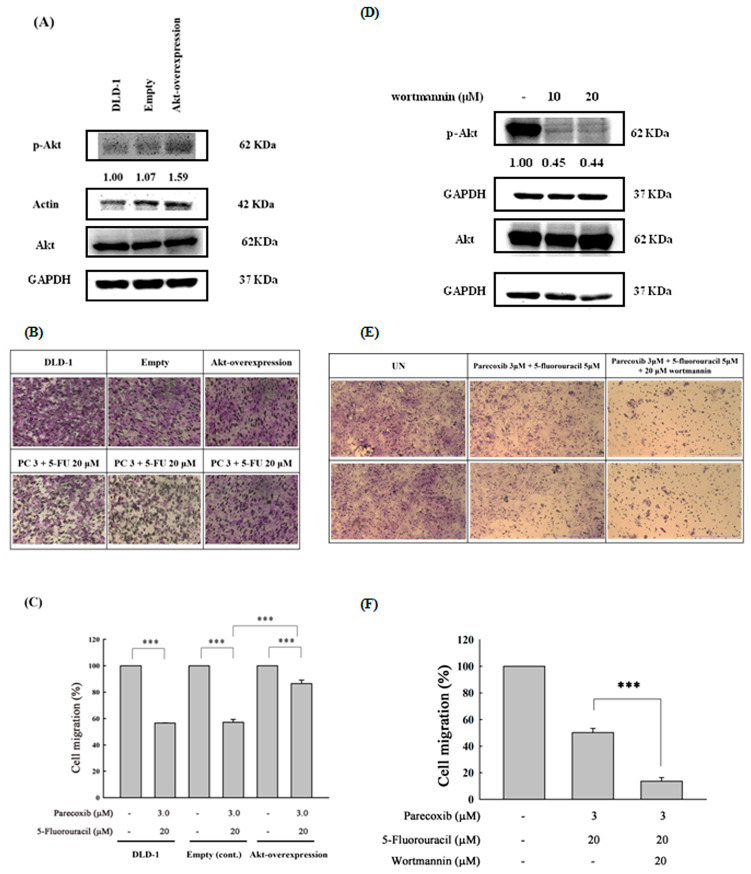
Effect of overexpression of Akt phosphorylation in parecoxib- and 5-FU-treated DLD-1 cells. (**A**,**D**) Phosphorylation of Akt was detected by Western blot. GAPDH and actin were selected as loading controls. (**B**,**E**) Random areas from each of the triplicate migration assays were assessed using a phase-contrast microscope (magnification 200×). (**C**,**F**) The absorbance of crystal violet was detected at 570 nm by using a microplate reader. The data are shown as the mean ± SD of separate tests. Significant differences are expressed as *p* < 0.001 (***).

**Figure 8 biomedicines-12-01526-f008:**
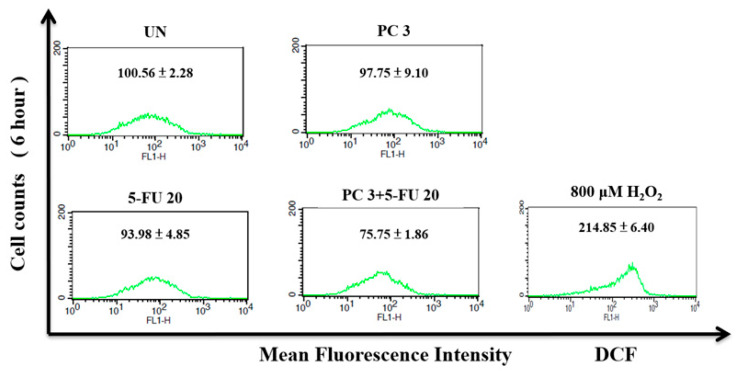
Effect of parecoxib and 5-FU on intracellular ROS in DLD-1 cells. After treatment, all cells were incubated with DCFH-DA for intracellular ROS detection and assessed using a flow cytometer. The data in each panel show the mean fluorescence intensity of DCF inside the cells. The data are shown as the mean ± SD (*n* = 5–8) of individual experiments.

**Figure 9 biomedicines-12-01526-f009:**
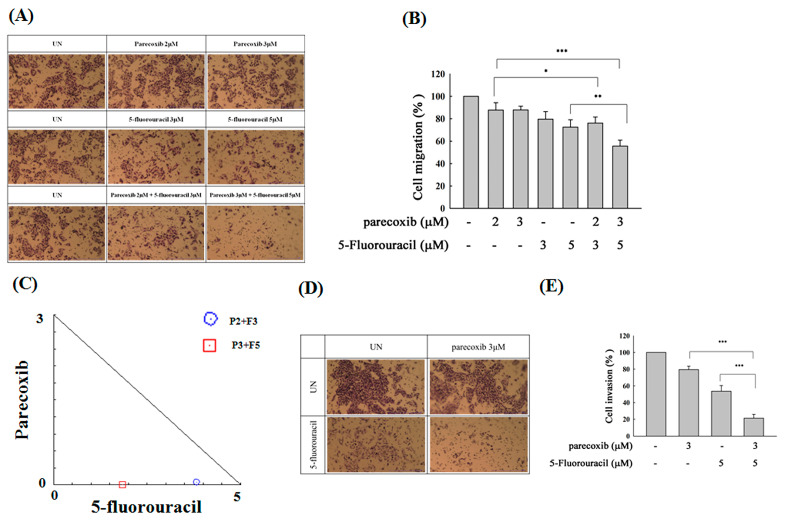
Effect of parecoxib and 5-FU on cell migration and invasion by transwell and matrix gel assays in SW480 cells. (**A**,**B**) Migration assay. (**D**,**E**) Invasion assay. (**A**,**D**) Random fields from each of the triplicate migration assays were calculated by a phase-contrast microscope (magnification 200×). (**B**,**E**) The absorbance of crystal violet was determined at 570 nm by using a microplate reader. The values are displayed as the mean ± SD of separate trials. Significant differences are expressed as *p* < 0.05 (*), *p* < 0.01 (**), and *p* < 0.001 (***). (**C**) Isobologram analysis of the parecoxib and 5-FU combination in SW480 cells. The trials were conducted at least three times. A descriptive trial is shown.

## Data Availability

The data that support the findings of this study are contained within the article.

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
