# Peer review of "Parecoxib and 5-Fluorouracil Synergistically Inhibit EMT and Subsequent Metastasis in Colorectal Cancer by Targeting PI3K/Akt/NF-κB Signaling"

_biomedicines, 2024, doi:10.3390/biomedicines12071526_

Round 1
Reviewer 1 Report
Comments and Suggestions for Authors
The article discusses an important topic. This study conducted in the laboratory has demonstrated that when parecoxib and 5-fluorouracil are used in appropriate doses, they can synergistically inhibit the metastasis of colorectal cancer cells. These findings are promising and should be validated in clinical practice. If proven effective, they have the potential to change colorectal cancer plan treatment.
However, the title, abstract, introduction, methods and results presentation, discussion, and references must be improved.
Title – only alludes to some potential mechanisms studied implicated in Parecoxib and 5-fluorouracil synergistic inhibition of metastasis of colorectal cancer cells.
Abstract – proper
Introduction – can be improved:
5-fluorouracil/capecitabin is a first-line drug used in colon cancer stage III or IV as systemic chemotherapy (adjuvant chemotherapy or a palliative, alone or in association with oxaliplatin. 5-fluorouracil is administrated intravenously. Capecitabin is administrated per os.
5-fluorouracil/capecitabin can also be used in rectal cancer as one of the drugs in neoadjuvant therapy and in total neoadjuvant therapy or as adjuvant therapy alone or in association with others.
Approximately 40% of CRC patients treated with 5-fluorouracil do not respond to the treatment or develop resistance. In such cases, 5-fluorouracil/capecitabin and oxaliplatin, which have a synergistic effect, is crucial. However, the use of Oxaliplatin is associated with several side effects. If a Cox-2 inhibitor could replace Oxaliplatin, it would significantly advance CRC treatment.
Celecoxib, a Cox-2 inhibitor drug, can be used in clinical practice in patients treating familial adenomatous polyposis and patients with extent colorectal polyposis. Parecoxib is also a Cox-2 inhibitor but is used intravenously.
NSAIDs are inhibitors of the cyclooxygenase enzyme family, which catalyzes the metabolism of arachidonic acid to prostaglandins, prostacyclin, and thromboxane. The cyclooxygenase-2 isoform is induced in response to cytokines and growth factors and is expressed in inflammatory disease, premalignant lesions (such as colorectal adenomas), and colon cancer.
It is important to clarify some of these concepts in the manuscript introduction.
Methods and results
See what is marked in the manuscript
Discussion
See what is marked in the manuscript.
The discussion must be focused in:
1 5-fluorouracil mechanism effect in colorectal cancer cells
2 Parecoxib mechanism effect in colorectal cancer cells
3 Hypostatized how 5-fluorouracil and Parecoxib can have an inhibitory synergistic effect on CRC tumor cell migration and invasion
4 How the results obtained in this paper confirm the previous items 1-3 and compared with the international literature available
5 If some results are unexpected, what are the reasons?
6 Analyze the weak and strong points of the methodology utilized and the results obtained.
7 How can we confirm these laboratory results in cell culture to establish a future connection between these basic science findings and clinical practice?
Conclusion
Proper
References
See what is marked in the manuscript.
Must be improved
Author Response
Response to Reviewer 1:
We sincerely thank you for your constructive and valuable comments that helped us improve our manuscript. Herein, we provide our point-by-point responses to the comments along with a description of all the changes that have been made and highlighted in blue in the revised manuscript.
The article discusses an important topic. This study conducted in the laboratory has demonstrated that when parecoxib and 5-fluorouracil are used in appropriate doses, they can synergistically inhibit the metastasis of colorectal cancer cells. These findings are promising and should be validated in clinical practice. If proven effective, they have the potential to change colorectal cancer plan treatment.
However, the title, abstract, introduction, methods and results presentation, discussion, and references must be improved.
Title – only alludes to some potential mechanisms studied implicated in Parecoxib and 5-fluorouracil synergistic inhibition of metastasis of colorectal cancer cells.
We modify the title to “Parecoxib and 5-fluorouracil synergistically inhibit EMT and subsequent metastasis in colorectal cancer by targeting PI3K/Akt/NF-κB signaling”. (Please see Title)
Abstract – proper
Introduction – can be improved:
5-fluorouracil/capecitabin is a first-line drug used in colon cancer stage III or IV as systemic chemotherapy (adjuvant chemotherapy or a palliative, alone or in association with oxaliplatin. 5-fluorouracil is administrated intravenously. Capecitabin is administrated per os.
5-fluorouracil/capecitabin can also be used in rectal cancer as one of the drugs in neoadjuvant therapy and in total neoadjuvant therapy or as adjuvant therapy alone or in association with others.
Approximately 40% of CRC patients treated with 5-fluorouracil do not respond to the treatment or develop resistance. In such cases, 5-fluorouracil/capecitabin and oxaliplatin, which have a synergistic effect, is crucial. However, the use of Oxaliplatin is associated with several side effects. If a Cox-2 inhibitor could replace Oxaliplatin, it would significantly advance CRC treatment.
Celecoxib, a Cox-2 inhibitor drug, can be used in clinical practice in patients treating familial adenomatous polyposis and patients with extent colorectal polyposis. Parecoxib is also a Cox-2 inhibitor but is used intravenously.
NSAIDs are inhibitors of the cyclooxygenase enzyme family, which catalyzes the metabolism of arachidonic acid to prostaglandins, prostacyclin, and thromboxane. The cyclooxygenase-2 isoform is induced in response to cytokines and growth factors and is expressed in inflammatory disease, premalignant lesions (such as colorectal adenomas), and colon cancer.
It is important to clarify some of these concepts in the manuscript introduction.
Thank you for your comments. We have clarified these concepts in the introduction section of our manuscript. (Please see the third and fourth paragraphs of Introduction section)
Methods and results
See what is marked in the manuscript
Thank you for your comments. We have revised the manuscript according your suggestions. (Please see Methods and Results sections)
Answer the question of Figure 2B:
Previous literature has shown that 5-FU inhibits migration and invasion of colorectal cancer cells through PI3K/Akt pathway [36]. Similarly, we confirmed 5-FU (15 μM and 20 μM) alone could inhibit about 30% of migration and the inhibition did not dose-dependent manner (Figure 2B). These results indicating that 5-FU alone can moderate inhibit migration and invasion of colorectal cancer cells. Our present results find that parecoxib (3 μM) slightly increased the migration but it is not significant. (Please see the fourth paragraph of Discussion section)
Answer the question of Figure 3B:
In Figure 3B, the inhibition of migration of 5-FU/parecoxib combination is weaker at 48 h than at 24 h. It may be that the cancer cells were adapting to the crawling environment between 0 and 24 h, making the inhibitory effect of migration on the drugs more significant. However, during 24 to 48 h, part of the drugs may be metabolized, so the inhibitory effect is relatively weak. (Please see the fourth paragraph of Discussion section)
Discussion
See what is marked in the manuscript.
Thank you for your comments. We have revised the manuscript according your suggestions. (Please see the Discussion section).
The discussion must be focused in:
- 5-fluorouracil mechanism effect in colorectal cancer cells
5-Fluorouracil (5-FU) represents an anti-metabolite with replacement of fluorine for hydrogen at the C-5 position of uracil. The thymine-uracil/5-FU exchange produced by thymine substitute in DNA therefore results in the formation of adenine-uracil/5-FU base pairs [33]. The mechanism effect of 5-FU on antitumor is chiefly via the repression of thymidylate synthase leading to disturbing the intracellular deoxynucleotide pools needed for DNA replication [33]. (Please see the first paragraph of Discussion section.)
- Parecoxib mechanism effect in colorectal cancer cells
Our previous study demonstrates the parecoxib anti-metastasis mechanism effect is correlated with the attenuated phosphorylation of Akt, expression of vimentin (a mesenchymal marker), and β-catenin, and corresponded with the upregulated GSK3β and E-cadherin (an epithelial marker) in human colon cancer cells [24]. (Please see the first paragraph of Discussion section.)
- Hypostatized how 5-fluorouracil and Parecoxib can have an inhibitory synergistic effect on CRC tumor cell migration and invasion
Recent studies have approved that Akt/NF-κB signaling are strongly associated with metastasis in colorectal tumor cells [34, 35]. Parecoxib has been demonstrated by us to inhibit the Akt phosphorylation and EMT in human colorectal cancer at clinical concentrations [24]. 5-FU has the ability to inhibit migration and invasion of CRC cells via PI3K/Akt pathway [36]. It is necessary to find a potential drug to enhance the anti-metastatic ability of 5-FU. Thus, we speculated that in colorectal cancer cells, the anti-migration and anti-invasion efficacy of 5-FU might be increased in combination with parecoxib. (Please see the first paragraph of Discussion section.)
- How the results obtained in this paper confirm the previous items 1-3 and compared with the international literature available
Our results demonstrated that parecoxib enhanced the inhibition of 5-FU on the EMT (Figure 4) and PI3K/Akt/NF-κB pathway (Figure 6 and 7) and subsequent enhanced the inhibition of MMP-9 expression and activity (Figure 5). These results confirm our hypothesis that 5-FU and parecoxib exhibit an inhibitory synergistic effect on migration and invasion in colorectal cancer cells. PI3K/AKT signaling pathway has been demonstrated to be a critical regulator of epithelial-mesenchymal transition in colorectal tumor cells [37]. There are many compounds and substances can exhibit anti-metastasis through inhibiting PI3K/AKT signaling pathway and regulating EMT. For example, berberine inhibited migration and invasion of some colon cancer cell lines through up-regulating PTEN which repressed the PI3K/AKT pathway at the gene and protein levels [38]. Antrodia camphorate significantly inhibited migration and invasion, accompanied by the down-regulation of MMP-2 and MMP-9 proteins via the inhibition of the PI3K/AKT/NFκB signaling pathways [39]. Astragalus mongholicus Bunge-Curcuma aromatica Salisb inhibited the expression and transcription of genes related to the PI3K/AKT pathway while inhibiting the EMT process in colon cancer cells and model mice [40]. Our present findings of this research are consistent with these previous studies. Inhibition of PI3K/Akt/NF-κB pathway, EMT and MMP-9 are critical events in anti-metastasis effect of 5-FU/parecoxib combination in colorectal cancer. (Please see the first paragraph of Discussion section.)
- If some results are unexpected, what are the reasons?
5-FU can increase ROS in colorectal cancer cells [54]. Our present study reveals that 5-FU/parecoxib combination did not increase the intracellular ROS levels. Conversely, 5-FU/parecoxib combination inhibited the intracellular ROS levels. The result is unexpected. COX-2 is an important source of ROS generation. COX-2 derived different prostanoids can modulate ROS production [55]. One possible reason is that parecoxib is an NSAID, which can specifically inhibit the production of ROS mediated by COX-2, resulting in the inability of 5-FU/parecoxib combination to induce an increase in ROS. (Please see the third paragraph of Discussion section.)
- Analyze the weak and strong points of the methodology utilized and the results obtained.
The weak point of the methodology is lack PCR method to analyze various genes expression. However, the PCR method cannot evaluate the phosphorylation of Akt. The strong points of the methodology are to use the Western blot to analyze various protein expressions including the phosphorylation of Akt and to use the zymography to analyze the MMP-9 activity. Moreover, the overexpression of p-Akt to confirm the detailed anti-metastasis mechanism in 5-FU/parecoxib combination is also a strong point of the methodology. (Please see the last second paragraph of Discussion section.)
- How can we confirm these laboratory results in cell culture to establish a future connection between these basic science findings and clinical practice?
Both 5-FU and parecoxib are already commonly used drugs in clinical practice. The concentrations used in our study were also within the human serum concentration range. Animal experiments of xenograft colorectal cancer can be planned in the future to prove the anti-metastatic results obtained from our cell experiments to connect between these basic science findings and clinical practice. (Please see the last paragraph of Discussion section.)
Conclusion
Proper
References
See what is marked in the manuscript.
We have revised the references according the suggestions. (Please see References section.)
Must be improved

Reviewer 2 Report
Comments and Suggestions for Authors
The authors present the research driven by a clear hypothesis that is grounded in their previous work in this field. They have previously explored the mechanism of action of parecoxib and through this research they attempted to further investigate its application potential in combination with a standard chemotherapeutic drug. They offer solid conclusions and demonstrate that parecoxib can synergistically enhance 5-FU to inhibit metastasis in DLD-1 cells. The paper is easy to follow, with clearly presented results.
Minor points:
Several subtitles should be reformulated (2.10, 3.1, 3.4, 3.5, 3.6)
I did not understand the choice of different controls for Western blot considering that the starting material is the same cell line under similar treatments: actin and GAPDH were used for different EMT markers and PI3K/Akt/NF-κB pathway, then tubulin for MMP, IKK and p65, while histone H3 is mentioned, but does not appear in Fig. 6
The Fig. 7 legend contains a few errors („Overexpression of Akt phosphorylation“, LDL-1 cells)
There are a few confusing sentences in the Discussion which should be reformulated: The concentration of ROS in the cell determines the role of regulation (Line 401), To discover whether Snail is included with the EMT course... (line 450)
Comments on the Quality of English LanguageNo major issues detected
Author Response
Response to Reviewer 2:
We sincerely thank you for your constructive and valuable comments that helped us improve our manuscript. Herein, we provide our point-by-point responses to the comments along with a description of all the changes that have been made and highlighted in blue in the revised manuscript.
The authors present the research driven by a clear hypothesis that is grounded in their previous work in this field. They have previously explored the mechanism of action of parecoxib and through this research they attempted to further investigate its application potential in combination with a standard chemotherapeutic drug. They offer solid conclusions and demonstrate that parecoxib can synergistically enhance 5-FU to inhibit metastasis in DLD-1 cells. The paper is easy to follow, with clearly presented results.
Minor points:
Several subtitles should be reformulated (2.10, 3.1, 3.4, 3.5, 3.6)
We have reformulated subtitles according your mentions as follow:
2.10. Effect of p-Akt Overexpression
3.1. Cell Viability on Parecoxib and 5-FU Treatment
3.4. EMT suppression on Parecoxib and 5-FU Combination
3.5. MMP-9 Inhibition on Parecoxib and 5-FU Combination
3.6. Inhibition of PI3K/Akt/NF-κB Pathway on Parecoxib and 5-FU Combination
I did not understand the choice of different controls for Western blot considering that the starting material is the same cell line under similar treatments: actin and GAPDH were used for different EMT markers and PI3K/Akt/NF-κB pathway, then tubulin for MMP, IKK and p65, while histone H3 is mentioned, but does not appear in Fig. 6
EMT markers and PI3K/Akt/NF-κB pathway were completed in different periods. Although we purchased the same catalog number of loading control antibodies for subsequent experiments, they could not show obvious Western blots, so we replaced them with different loading control antibodies. We delet the histone H3 in Fig.6 legend.
The Fig. 7 legend contains a few errors („Overexpression of Akt phosphorylation“, LDL-1 cells)
We have revised the Fig. 7 legend to eliminate these errors. (Please see Fig. 7 legend)
There are a few confusing sentences in the Discussion which should be reformulated: The concentration of ROS in the cell determines the role of regulation (Line 401), To discover whether Snail is included with the EMT course... (line 450)
We revised the sentences as follow:
“The intracellular ROS concentration determines the direction of dual role.” (Please see page 15, line 445)
“To discover whether Snail is included with the EMT process,” (Please see page 16, line 513-514)
